# Bioactive compounds, antioxidant capacity and anti-inflammatory activity of native fruits from Brazil

**Bruna Tischer[1,2], Philipus Pangloli[1], Andrea Nieto-Veloza[1¤], Matthew Reeder[1], Vermont P. Dia[1]***

**1** Department of Food Science, The University of Tennessee Institute of Agriculture, Knoxville, Tennessee, United States of America, **2** Institute of Food Science and Technology, Federal University of Rio Grande do Sul, Porto Alegre, Rio Grande do Sul, Brazil

¤ Current address: Department of Food Engineering, Fundacion Universitaria Agraria de Colombia – UNIAGRARIA, Bogota, Colombia

* vdia@utk.edu

**Data Availability Statement:** All relevant data are within the manuscript and its Supporting information files.

## Abstract

The purpose of this study was to extract, identify, and quantify the phenolic compounds in grumixama (*Eugenia brasilienses* Lam.) and guabiju (*Myrcianthes pungens*), native fruits from southern region of Brazil, and to explore their antioxidant and anti-inflammatory properties. The phenolic compounds were extracted with acidified water and acidified methanol and evaluated for their bioactive constituents, antioxidant capacity, and anti-inflammatory properties. Spectrophotometric quantification shows tannins to be the most prevalent at 2.3 to 5.8 g/100g fresh fruit with acidified methanol containing higher concentrations of different phenolics than acidified water. HPLC analysis indicates that gallic acid, catechin, vanillic acid, and ellagic acid are the most prevalent phenolics in the two fruits extracts. Scavenging of DPPH and NO radicals showed inhibition by as much as 95% and 80%, respectively, at 2.5 gallic acid equivalent (GAE)/mL of the extract. At 50 μg GAE/mL, the release of pro-inflammatory molecules NO and IL-6 was significantly reduced with acidified methanol extract having higher inhibitory activity. Our results revealed that these native fruits, grown in the south of Brazil, are rich sources of phenolic compounds and have great antioxidant and anti-inflammatory activity.

## Introduction

Brazil has a large number of native fruits distributed throughout the country. Many of them have very interesting pharmacological and nutritional properties that could be used for food and medicinal purposes [1]. Unfortunately, many of these fruits are not well known, therefore they are poorly studied or consumed. The lack of consumer interest, the low economic value for commercial production, and the limited interest and effort to perform research and development with these fruits, has brought some of them to the path of extinction [2].

Some studies have shown several promising pharmacological properties such as antioxidant, anti-inflammatory, antidiabetic, anticancer, antiviral, and antitumoral activities,

**Funding:** Fulbright Commission for the research grant granted and for providing the opportunity for the partnership with the University of Tennessee. This study is partially funded by HATCH TEN00585 to VPD. The funders had no role in study design, data collection and analysis, decision to publish, or preparation of the manuscript.

**Competing interests:** The authors have declared that no competing interests exist.

indicating the high potential of these fruits to protect the human body from harmful and degenerative biological processes, and to decrease the risk of chronic diseases [3]. These pharmacological activities have been associated with phenolics, terpenes, alkaloids, and other biologically active compounds present in the fruits [4]. In addition, the presence of chemical structures that could serve to replace different synthetic additives currently used in foods, have been reported in these Brazilian native fruits [5, 6].

Among these fruits are grumixama (*Eugenia brasilienses* Lam.) and guabiju (*Myrcianthes pungens*), native fruits to Rio Grande do Sul that are good sources of bioactive compounds with potential benefits for human health [7–9]. Grumixama belongs to the *Myrtaceas* family, and it is distributed along the south and southeast regions of Brazil. It has small cherry fruits of about 2 cm in diameter, which generally contain 1 to 3 seeds, and the pulp is juicy and firm with a sweet acidic flavor [9]. Guabiju is a spherical velvety fruit, with a purplish color, and a succulent yellowish edible pulp when ripe [8]. Despite these fruits are commercially available as frozen fruits, frozen pulp, or jelly, they are not currently used for food production at large scale but rather restricted to handcraft production. Few research works, and mainly focused on the characterization of the fruits, have been done on grumixama and guabiju [10–16]. Several researchers suggest that more studies using cell and animal models, as well as human preclinical and clinical trials, are necessary to support the potential biological functionality of these fruits [9, 17].

Thus, this study aimed to characterize the phenolic compounds present in grumixama and guabiju, and to study their biological activities including *in vitro* antioxidant properties, and anti-inflammatory capabilities using activated murine immune cells as a model for inflammation. Phenolic compounds were both quantified spectrophotometrically and individual compounds were identified and quantified by high performance liquid chromatorgraphy (HPLC) using different analytical standards. These standards were chosen based on availability as well as those reported in the literature to be present in these fruits.

## Materials and methods

### Samples and materials

Grumixama (purple or α-variety) and guabiju fruits were collected in Guaíba (Latitude: -30.1141, Longitude: -51.3281 30˚ 6′ 51′′ South, 51˚ 19′ 41′′ West) and Venâncio Aires city (Latitude: -29,606, Longitude: -52.1944 29˚ 36′ 22′′ South, 52˚ 11′ 40′′ West), Rio Grande do Sul, Brazil. The fruits were thoroughly washed with tap water, frozen at -70˚C for 24 h, lyophilized (L101, Liobras, Brazil), and vacuum packed for long term storage. Moisture content was determined before and after lyophilization, initial and final moisture were 80.3±1.7 and 15.1±0.6 for grumixama, respectively, and 78.7±1.2 and 15.3 ±0.4 for guabiju. All materials and reagents were purchased from the following suppliers unless otherwise specified: Gels were obtained from GenScipt USA Inc. (Piscataway, NJ, USA), murine macrophages RAW 264.7 cell line was from the American Type Culture Collection (ATCC, Manassas, VA, USA), cell culture media, supplements and other cell-work related materials and reagents were purchased from Life Tech (Carlsbad, CA, USA) and Corning Inc. (Corning, NY, USA); all other chemicals were purchased from either Fisher Scientific (Hampton, NH, USA) or Sigma-Aldrich (Saint Louis, MO, USA). Absorbance of reaction mixtures in 96-well plates was read using a Cambrex ELX808 microplate reader (Biotek Instruments, Winooski, VT, USA) unless something different is specified.

### Extraction of phenolics

Phenolic compounds were extracted from grumixama and guabiju using acidified water and methanol (0.1% HCl, v/v) according our previous work with some modifications [17]. Briefly,

lyophilized grumixama and guabiju fruits were ground in a mortar with pestle and passed through a 30-mesh sieve. Approximately 20 g of each ground fruit was suspended in 200 mL of acidified water or acidified methanol, stirred continuously for 16 h at room temperature in the dark, and then centrifuged at 8000 × g for 20 min at 4 ℃ (Sorvall LYNX 6000, ThermoFisher, Waltham, MA, USA). The supernatant was collected as phenolics enriched extract. A 10 mL aliquot of each extract was sampled and stored at 4 ℃ in the dark until analysis of phenolic compounds (total polyphenol, total 3-deoxyanthocyanidins, total monomeric anthocyanin, total flavonoids, and total tannins). Methanol extracts were concentrated by rotary evaporation (Büchi® Rotovapor R-200, Buchi Corp, New Castle, DE, USA) under vacuum at 50 ℃. The remaining liquid in the methanol and water extracts were freeze dried (FreeZone Triad Freeze Dryers, Labconco, Kansas City, MO, USA). The dried extracts were stored in amber glass bottles under a moisture-controlled environment until use for identification of phenolic compounds and measurement of antioxidant and anti-inflammatory properties. Each extraction was performed in triplicate for each fruit.

## Spectrophotometric quantification of total phenolic compounds

**Total polyphenol concentration by Folin-Ciocalteu method.** Total polyphenols were determined by Folin-Ciocalteu method according to a previous protocol [17]. Ten μL of sample extracts (water and methanol), and gallic acid standards (ranging from 0 to 1000 μg/mL) were plated in triplicates into a 96-well plate. Then, 25 μL of 1N Folin-Ciocalteu reagent, 25 μL of 20% sodium bicarbonate, and 150 μL of deionized water were added. After incubation at room temperature for 30 min in the dark, absorbance was read at 630 nm. The concentration of total polyphenols was calculated using the gallic standard curve and expressed as mg gallic acid/100 g fresh fruit.

**Total tannins content by HCl-vanillin method.** Total tannins were quantified using a previously reported protocol [17]. Briefly, 20 μL of sample extracts and catechin standards (0–1000 μg/mL) were loaded in triplicate into a 96-well plate, followed by 30 μL of methanol and 150 μL of vanillin working reagent. Vanillin working reagent was prepared by mixing equal volumes of 1% vanillin and 8% HCl (v/v) in methanol. After incubation for 10 min at room temperature in the dark, absorbance was read at 490 nm. The total tannins were calculated using the catechin standard curve and expressed as mg catechin/100 g fresh fruit.

**Total flavonoids content by aluminum conjugation method.** Quantification of total flavonoids in the fruit extracts was performed according to our previous work [18]. Twenty μL of extracts and quercetin standard solutions (0–200 μg/mL) were plated in triplicate into a 96-well plate. Then 80 μL of methanol and 100 μL of 2% $AlCl_3.6H_2O$ (prepared in methanol) were added. After incubation for 30 min at room temperature in the dark, the absorbance was read at 405 nm. Total flavonoids were calculated using the quercetin standard curve and expressed in mg quercetin/100 g fresh fruit.

**Total anthocyanin and 3-deoxanthocyanins content.** Total anthocyanin and total 3-deoxanthocyanins were measured according to our previous work [17]. Briefly, 200 μL of sample extracts were plated in triplicate in a 96-well plate and the absorbance was read at 490, 520, and 700 nm using a Synergy HT microplate reader (Biotek Instrument, Winooski, VT, USA). The total anthocyanins (TA) and total 3-deoxanthocyanins (T-3DA) contents were calculated according to Eqs 1 and 2 and expressed as mg cyanidin-3-glucoside (C3G) or mg luteolinidin per 100 g fresh fruit, respectively.

$$TA \left( \frac{mg\ C3G}{100\ mg\ fresh\ fruit} \right) = \frac{(Abs520 - Abs700) * 449.38 * 1E6 * (vol,\ L)}{26,900 * 0.45 * (sample\ weight,\ g)} \tag{1}$$

$$T - 3DA \left( \frac{mg \ luteolinidin}{100 \ mg \ fresh \ fruit} \right) = \frac{(Abs490 - Abs700) * 271.24 * (vol, \ L)}{35,000 * 0.45 * (sample \ weight, \ g)} \qquad (2)$$

where 449.38 and 26,900 are the molecular weight and molar extinction coefficient for C3G, respectively; 271.24 and 35,000 correspond to the same parameters for luteolinidin, *vol* represents the volume of extract in liters, and 0.45 is the conversion factor from a conventional 1-cm pathlength method.

## Antioxidant capacity analysis

For antioxidant analyses, the lyophilized extracts were prepared as follows. The lyophilized extract was diluted in water (1:10), sonicated in water bath for 15 min, vortexed at 3,000 rpm for 60 min, centrifuged at 20,000 x g for 30 min, and filtered with 0.22 μm polyvinylidene (PVDF) membrane. The filtered supernatant was used to measure the antioxidant and anti-inflammatory properties. Measurements of antioxidant properties including 2,2-diphenyl-1picrylhydrazyl (DPPH), nitric oxide scavenging capacity (NO), and oxygen radical absorbance capacity (ORAC) assays were performed in triplicate following procedures previously published [17] with minor modifications.

**DPPH method.** Each filtered supernatant was diluted with deionized water to concentrations of 0 (blank), 2.5, 10.0, 20.0, and 40.0 and μg gallic acid equivalents (GAE)/mL and 100 μL of each concentration were loaded twice in triplicate into a 96-well. Then 100 μL of methanol were added to one set of samples, while 100 μL of DPPH solution (100 μM freshly prepared in methanol) were added to the second set of samples. The microplate was incubated for 30 min at room temperature in the dark, and the absorbance was read at 517 nm using a Synergy HT microplate reader (Biotek Instruments). Results are presented as percentage of DPPH radicals produced relative to the blank according to Eq 3.

$$\%DPPH = \frac{Abs. \ of \ sample \ with \ DPPH - abs. \ of \ sample \ with \ methanol}{Abs. \ of \ blank} * 100\% \qquad (3)$$

**Deactivation of the peroxyl radical—ORAC method.** For ORAC analysis, 75 mM potassium phosphate buffer (pH 7.4) was used to prepare Trolox (6-hydroxy-2,5,7,8-tetramethyl-chroman-2-carboxylic acid) standards ranging from 0–100 μM, and to dilute the filtered supernatant to a final concentration of 50 μg GAE/mL. After which, 25 μL of samples and standards were loaded into a black 96-well plate, followed by 150 μL of sodium fluorescein working solution (81 nM fluorescein in phosphate buffer). The plate was incubated in the dark for 30 min at 37°C followed by addition of 25 μL of 2,2'-azobis (2-amidinopropane) dihydrochloride solution (152 mM in the phosphate buffer). A fluorescence spectrophotometer (Synergy microplate reader) was used to monitor fluorescence decay at 37°C, 485/20 nm excitation and 528/20 nm emission wavelengths, every one min for 120 min or until less than 0.5% of the initial value was reached. After subtraction of the blank, the area under the curve (AUC) for the standard curve and the samples were used for calculations. The results were expressed as μmol of Trolox per g fresh fruit (μmol TE/ g sample).

**NO scavenging method.** Fifty microliters of the filtered supernatant samples and control were pipetted into two separate 96-well plates, followed by 50 μL of distilled water. Then 25 μL of 100 mM sodium nitroprusside were added to the samples and controls in one plate, while 25 μL of DI water were added to the other. The plates were incubated at room temperature for 120 min, and then 100 μL of Griess reagent (prepared with equal volumes of 1% sulfanilic acid

in 5% phosphoric acid and 0.1% of N-(1-naphthyl)-ethylenediamine dihydrochloride) were added. The plates were further incubated for 15 min, and the absorbance was read at 550 nm using a Synergy HT microplate reader. For data analysis, the absorbance of the plate treated with DI water was subtracted from the absorbance of the plate treated with sodium nitroprusside. The results were presented as percentage NO production relative to control.

## Identification and quantification of phenolic compounds by HPLC analysis

Phenolic compounds in grumixama and guabiju fruits were identified and quantified according to a previous method [18] using Agilent 1200 HPLC system (Agilent Technologies, Santa Clara, CA), equipped with an auto-sampler (G1329A), a quaternary pump (G1311A), a UV diode array detector (G1315D), a degasser (G1322A), and a column thermostat (G1316A). The compounds were separated on a Zorbax Eclipse C-18 column ($4.6 \times 150$ mm, 5.0 μm; Agilent Technologies), maintained at 35°C. The injection volume was 20 μL. The mobile phase consisted of 4% formic acid in HPLC water (solvent A) and acetonitrile (solvent B) and the flow rate was 1 mL/min with gradient system as follows: 0–20 min, 12–20% B; 20–40 min, 20–50% B; 40–50 min, 50% B; 50–52 min, 50–12% B; and 52–55 min, 12% B. The detector was set at 280 and 340 nm. Firstly, 100 mg of the freeze-dried extracts were diluted in 1 mL of deionized water, sonicated for 15 min, vortexed for 60 min, and centrifuged at $20,000 \times g$ for 60 min. The supernatant was filtered with a 0.45 μm PVDF membrane and used for HPLC analysis. The final concentrations of samples were diluted accordingly, so that the absorbances were within the linear for the respective standard curves. Standard curves were prepared using commercial phenolic compounds including gallic acid, catechin, vanillic acid, coumaric acid, ellagic acid, naringenin, caffeic acid, ferulic acid, sinapic acid, quercetin, luteolin, apigenin, chrysin, syringic acid, and eriodicytol. Phenolic standards were diluted in acetonitrile to final concentrations of 0–100 μg/mL. The concentrations of phenolic compounds were expressed as mg of the specific compound per g of fresh fruit.

## Anti-inflammatory activity

**Cell culture and treatment.** RAW 264.7 murine macrophages activated with lipopolysaccharide (LPS) were used as an *in vitro* model of inflammation following the methods described by [19, 20]. Cells were grown in DMEM media supplemented with 10% heat-inactivated FBS and 1% of penicillin/streptomycin at 37 ºC in a humidified 5% $CO_2$ incubator. Cells were monitored under the microscope regularly and sub-cultured every two days or when reaching 80% confluency. Grumixama and guabiju extracts prepared in Section 2.4 were diluted with growth media to obtain the desired concentrations. For anti-inflammatory activity experiments, $2.5 \times 10^4$ cells per well were seeded in a 96-well plate with 200 μL of growth media and allowed to attach overnight. Then, the media was removed and replaced with fresh growth media alone (negative control), or containing 1 μg/mL LPS (positive control) or 1 μg/mL LPS + fruits extracts at final concentration of 50 and 100 μg GAE/mL (treatments). After 24 h incubation at 37 ºC, the media was carefully collected and stored at -20 ºC for further analysis. Cell viability was evaluated by spectrophotometry using MTS assay (Promega, Madison, WI, USA) following manufacturer's protocol; empty wells (without cells) treated with the appropriate media, were used as blanks to account for background signal. After subtracting the background signal, cell viability was calculated as % of viable cells relative to the negative control.

**Nitric oxide release assay.** Concentration of nitric oxide in the extracellular media was measured using Griess assay. Sodium nitrite standard solutions ranging from 0–200 μM were prepared using growth media. One hundred μL of each standard concentration, and of media collected in Section 2.6.1 were plated in triplicate in a 96-well plate. Then, 100 μL of Griess

reagent, prepared as described in section 2.4.3, were added. The plate was incubated at room temperature for 5 min and the absorbance was read at 550 nm. The concentration of NO was calculated using the sodium nitrite standard curve.

**Secreted IL-6.** The concentration of the pro-inflammatory cytokine IL-6 in the extracellular environment (media collected in section 2.6.1), was measured via ELISA using the commercial ELISA MAX™ Deluxe kit (BioLegend, San Diego, CA, USA), according to the manufacturer's instructions. The results were calculated using a standard curve and expressed in IL-6 (% production relative to positive control, LPS+).

## Statistical analysis

Results are presented as mean±standard deviation, and t-student test was used to establish significant differences between the extracts using acidified water or acidified methanol for each fruit with a confidence level of 95%. For the anti-inflammatory properties, analysis of variance (ANOVA) followed by the posthoc test Tukey were used to identify significant differences between the different concentrations of the same extract (p-value < 0.05).

## Results

### Phenolics quantification by spectroscopy

Quantification of phenolic compounds extracted with acidified water and acidified methanol from grumixama and guabiju are shown in Table 1. In general, the amounts of phenolic compounds extracted with acidified methanol were significantly higher than those extracted with acidified water suggesting that methanol is a more suitable extraction solvent for phenolic compounds than water. Among the different phenolics tested, tannins are the most prevalent group, being present in both fruits at the level of 2.3 to 5.8 g/100g fresh fruit, followed by total polyphenols at 248–547 mg/100 fresh fruit. Although the concentration of all compounds seems to be comparable between the two fruits, it is apparent that the concentration of polyphenols is higher in grumixama, while tannins are considerably more concentrated in guabiju.

### Phenolics identification and quantification by HPLC

The specific compositions of phenolic compounds present in grumixama and guabiju extracts were further studied using HPLC. Fig 1 shows the chromatographic profile of grumixama and guabiju extracts in water and methanol. Out of the 15 phenolic compounds investigated, only 11 were identified in the fruit extracts (Table 2). No peaks associated with apigenin, chrysin, syringic acid, or eriodyctiol were identified in any of the extracts, suggesting that these specific

**Table 1. Phenolic compounds in grumixama and guabiju extracts obtained with different solvents.**

| Phenolics | Grumixama (acidified water) | Grumixama (acidified methanol) | Guabijú (acidified water) | Guabijú (acidified methanol) |
|---|---|---|---|---|
| Total polyphenols (mg Gallic acid/100 g fresh fruit) | 247.6±1.1[b] | 546.6±14.9[a] | 285.5±5.7[b] | 399.98±15.8[a] |
| Total tannins (mg catechin/100 g fresh fruit) | 2344.7±193.7[b] | 3929.0±149.4[a] | 2933.8±183.0 [b] | 5846.9±194.0 [a] |
| Total anthocyanin (mg cyanidin-3-glucoside eq/100 g fresh fruit) | 63.5±0.4[b] | 108.2±0.3[a] | 99.6±11.7[b] | 140.2±16.2[a] |
| Total 3-deoxyanthocyanidins (mg luteonidin/100 g fresh fruit) | 29.5±0.6[b] | 55.7±1.5[a] | 47.8±3.8[b] | 67.7±4.6[a] |
| Total flavonoids (mg quercetin/100 g fresh fruit) | 43.0±0.9[b] | 50.9±1.7[a] | 39.9±0.8[b] | 44.8±0.8[a] |

Statistical treatment was applied to each fruit separately, comparing only the solvent used for each fruit. Quantitative comparison tests between fruits were not performed. Means with different letters within a row are significantly different (p < 0.05).

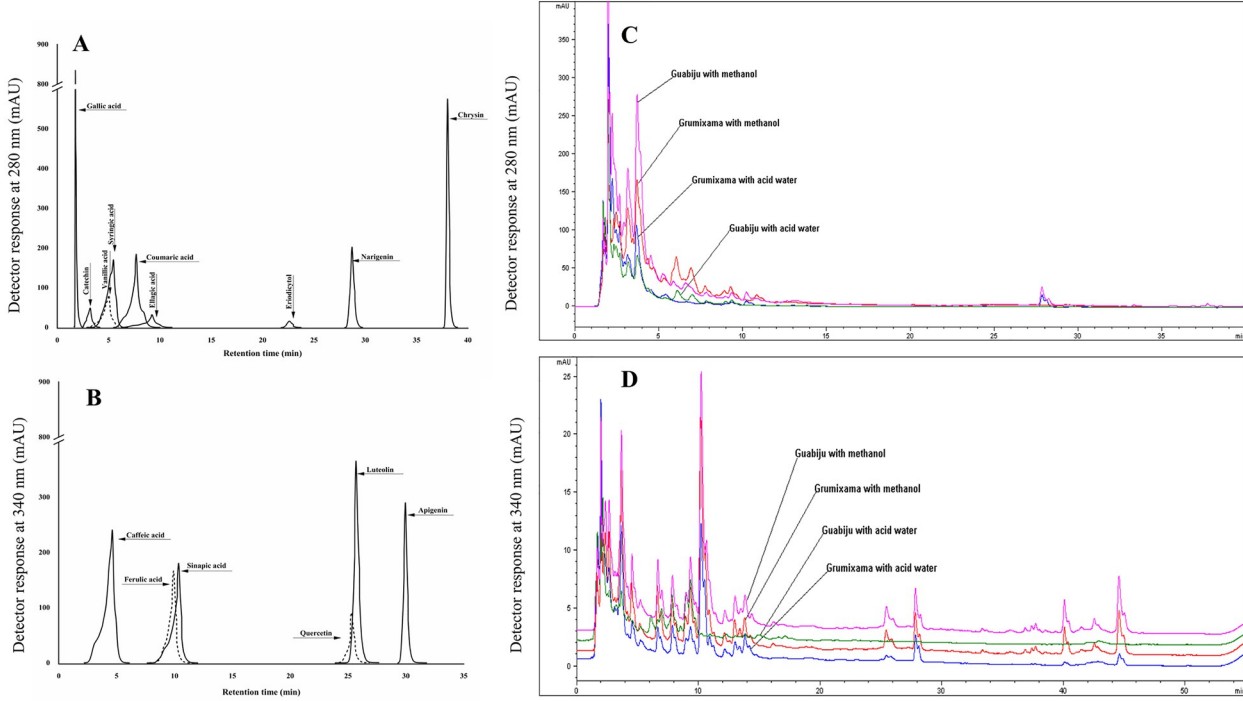

**Fig 1. Chromatographic profile of phenolics compounds in grumixama and guabiju extracted in acidified water or methanol.** *Chromatograms indicating the specific peak of phenolic standards detected at 280 nm (A) and 340 nm (B); chromatograms of grumixama (C) and guabiju (D) freeze-dried extracts at 100 mg mL⁻¹. Aamples at 12.5 mg/mL were injected for quantification of gallic acid and catechin, shown in Figs C and D, is an amplification of the first 5 minutes.

phenolic compounds might be absent in the studied samples. The concentration of all phenolic compounds identified in methanol extracts were significantly higher than those in acidified water extract, which is consistent with our findings regarding the total phenolic compounds (Table 1). Gallic acid, catechin, vanillic acid, and ellagic acid were identified as the phenolic compounds present in larger quantities in both fruits. Luteolin was not detected in grumixama while naringenin was not detected in guabiju.

**Table 2. Phenolic compounds and their levels (mg/100 g fresh fruit) in Grumixama and Guabiju extracts obtained with different solvents.**

| Phenolic compounds | WL (nm) | Grumixama (acidified water) | Grumixama (acidified methanol) | Guabijú (acidified water) | Guabijú (acidified methanol) |
|---|---|---|---|---|---|
| Gallic acid | 280 | 82.51±4.26[b] | 90.84±1.75[a] | 61.90±3.26[b] | 101.1±1.60[a] |
| Catechin | 280 | 292.23±0.67[b] | 572.32±9.15[a] | 205.90±3.51[b] | 485.10±1.29[a] |
| Vanillic acid | 280 | 29.21±1.64[b] | 42.20±5.64[a] | 41.09±0.91[b] | 76.60±4.80[a] |
| Coumaric acid | 280 | 3.30±0.05[b] | 12.20±1.87[a] | 16.77±0.57[b] | 64.82±0.74[a] |
| Ellagic acid | 280 | 32.31±1.32[b] | 58.58±3.27[a] | 34.32±8.25[b] | 70.35±7.87[a] |
| Naringenin | 280 | 7.91±0.15[b] | 9.42±0.19[a] | ND | ND |
| Caffeic acid | 340 | 2.82±0.04[a] | 3.68±0.01[a] | 14.87±0.05[b] | 24.77±1.14[a] |
| Ferulic acid | 340 | 3.14±0.06[a] | 4.15±0.07[a] | 23.90±1.37[a] | 29.07±1.74[a] |
| Sinapic acid | 340 | 4.00±0.04[b] | 6.33±0.12[a] | 1.74±0.02[a] | 2.22±0.03[a] |
| Quercetin | 340 | 0.55±0.01[b] | 0.91±0.02[a] | ND | 3.70±0.02 |
| Luteolin | 340 | ND | ND | ND | 1.94±0.03 |

Means in a row with different letters were significantly different ($p < 0.05$). Statistical treatment was performed for different solvents observing the same fruit.

## Antioxidant activity

Different antioxidant capabilities of grumixama and guabiju extracts were investigated. DPPH and NO scavenging capacity at different concentrations are presented in Table 3. No significant differences ($p > 0.05$) were observed in the % of DPPH production for the two extraction solvents in any of the fruits. Overall, the trend indicated that increasing concentrations of the extracts resulted in a lower level of DPPH production, and that both fruits were equally effective in scavenging the radicals. The results suggest that grumixama and guabiju possess excellent antioxidant activity, as the lowest concentration (2.5 µg GAE/mL) inhibited almost 95% of the DPPH radicals in the medium. It is worth mentioning that this concentration corresponds to less than 1 g of fresh fruit. In addition, the capability of grumixama and guabiju extract to scavenge DPPH radicals is much higher than that one reported for sorghum phenolics extracts [17].

Grumixama and guabiju extracts were also effective in scavenging NO radicals (Table 3). At the lowest concentrations (2.5 µg GAE/ml) the remaining NO radicals ranged from 19.6 to 27.3%. Thus, very small amounts of fresh fruit (less than 1 g) could inhibit 72.7–80.4% of the NO radicals. The effectiveness of the extracts also increased as the extracts concentrations increased, but the observed trend suggest that the capability of the grumixama extracts in scavenging NO radicals was higher than that of guabiju extracts.

The deactivation of peroxyl radical capacity by the ORAC method is presented in Fig 2. The ORAC values of 64.2 and 63.4 mmol Trolox equivalent/g fresh fruit for grumixama and guabiju methanolic extracts, respectively, were significantly higher ($P < 0.05$) than those found for the aqueous extracts, indicating an important effect of the extraction solvent. No significant differences were observed between the two fruits.

## Anti-inflammatory activity

To identify the potential anti-inflammatory activity of the extracts, mouse-derived innate immune cells (macrophages) were activated with LPS (LPS+) to induce a state of inflammation in the presence of two different concentrations of grumixama and guabiju extracts. Non-activated cells (LPS-) were used as control. The effect over cells viability is presented in Fig 3A. It

**Table 3. DPPH and NO scavenging activity of grumixama and guabiju extracts.**

| Extract Concentration* | Grumixama (acidified water) | Grumixama (acidified methanol) | Guabiju (acidified water) | Guabiju (acidified methanol) |
|---|---|---|---|---|
| **DPPH scavenging** | | | | |
| 2.5 µg GAE/mL | 5.54±0.08 | 5.85±0.92 | 5.43±0.56 | 5.57±0.24 |
| 10 µg GAE/mL | 4.02±0.03 | 4.01±0.24 | 4.50±0.38 | 4.30±0.02 |
| 20 µg GAE/mL | 3.16±0.80 | 3.73±0.02 | 3.86±0.20 | 4.02±0.08 |
| 40 µg GAE/mL | 2.82±0.32 | 2.89±0.24 | 3.24±0.28 | 3.32±0.08 |
| **NO scavenging** | | | | |
| 2.5 µg GAE/mL | 19.6±1.4 | 20.3±0.8 | 26.9±0.9 | 27.3±0.7 |
| 10 µg GAE/mL | 16.9±0.1 | 17.3±0.6 | 24.8±0.4 | 24.3±0.6 |
| 20 µg GAE/mL | 16.2±0.4 | 16.5±0.3 | 24.0±0.3 | 22.8±0.9 |
| 40 µg GAE/mL | 13.9±0.6 | 15.1±0.7 | 19.8±0.7 | 18.3±0.9 |

* GAE: Gallic acid equivalent.

For DPPH results are presents as % DPPH production relative to the blank (100%) and for NO results are presents as % sodium nitrite production relative to the blank (100%). Mean values within a row with different letter are significantly different ($p < 0.05$). No significant difference ($p > 0.05$) was found between the extracts using acidified water and acidified methanol for the same fruit, already increasing the concentration of the extract there is a significant difference between all responses, for the same solvent (same column).

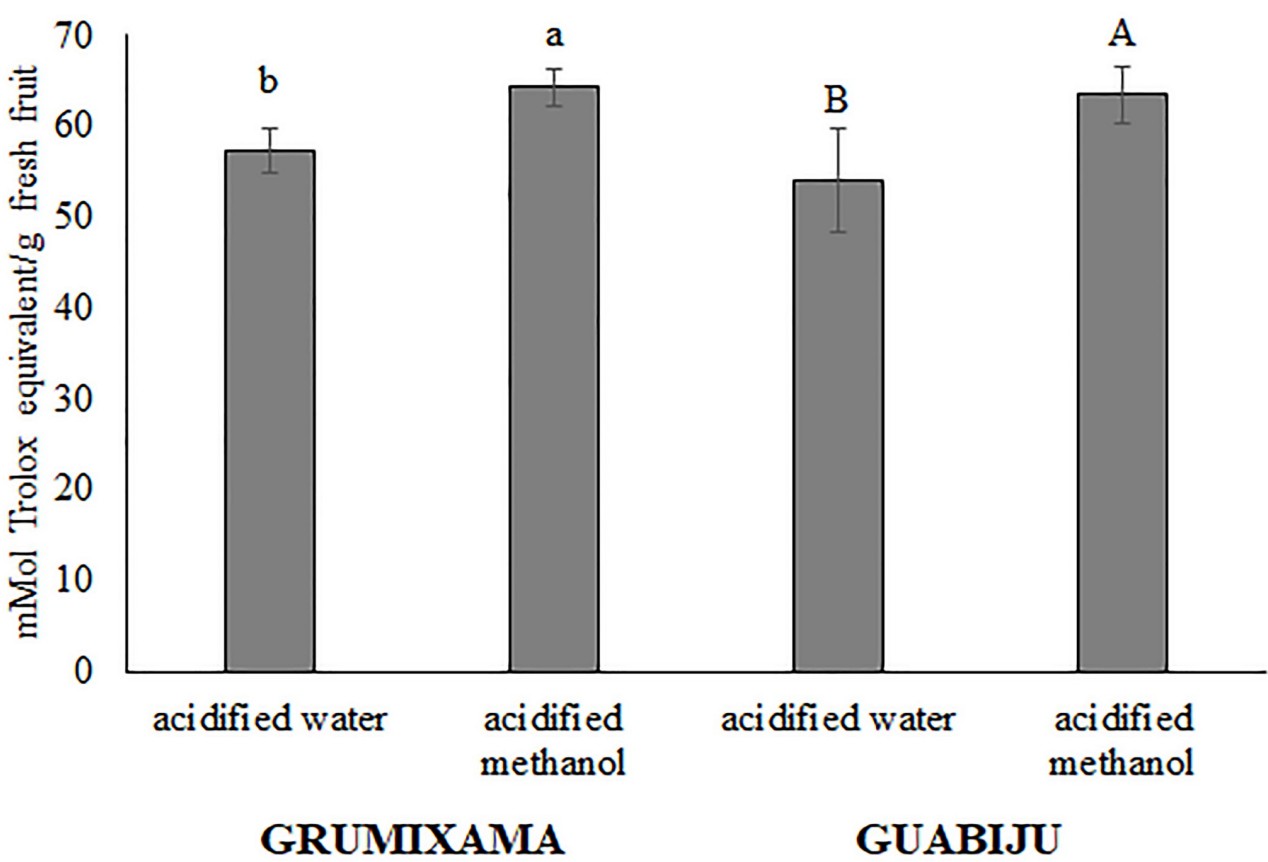

**Fig 2. Oxygen radical absorbance capacity (ORAC) for grumixama and guabiju extracts obtained with different solvents.** Different letters above columns indicate significant differences (p<0.05) between the treatments using the same extract for the same fruit extract.

is observed that none of the extracts at the lowest concentration (50 μg GAE/mL) had a significant effect over the cells viability when compared to non-activated cells. The difference observed with activated cells is due to the slight but non-significant increase in viability (2 to 3%) induced by LPS treatment. However, extracts at the highest concentration (100 μg GAE/mL) significantly decreased cells viability between 10 and 16%, suggesting that at high concentrations guabiju and grumixama extracts might exert a potential detrimental effect over immune cells in inflammatory state. No significant differences were found between the water and methanol extracts, except for guabiju at 100 μg GAE/mL, for which the methanolic extract exhibited a stronger effect over the cells viability when compared to the water extract at the same concentration.

Concentration of nitric oxide in the extracellular media is presented in Fig 3B. While non-stimulated cells (LPS-) produced negligible amounts of NO, LPS stimulation (LPS+) induced the production and release of a significant amount of this compound. The treatment with the extracts significantly reduced the release of NO between 50% and 65%, with the methanolic extracts exhibiting a statistically higher inhibitory effect than the water extracts in all the cases. Interestingly, for both fruits, the level of inhibition achieved by the two tested concentrations was statistically the same.

The release of the pro-inflammatory cytokine IL-6 is presented in Fig 3C. The increased concentration of IL-6 in the extracellular media, induced by LPS activation, was significantly reduced by both extracts in different proportions. Grumixama extract exhibited a dose

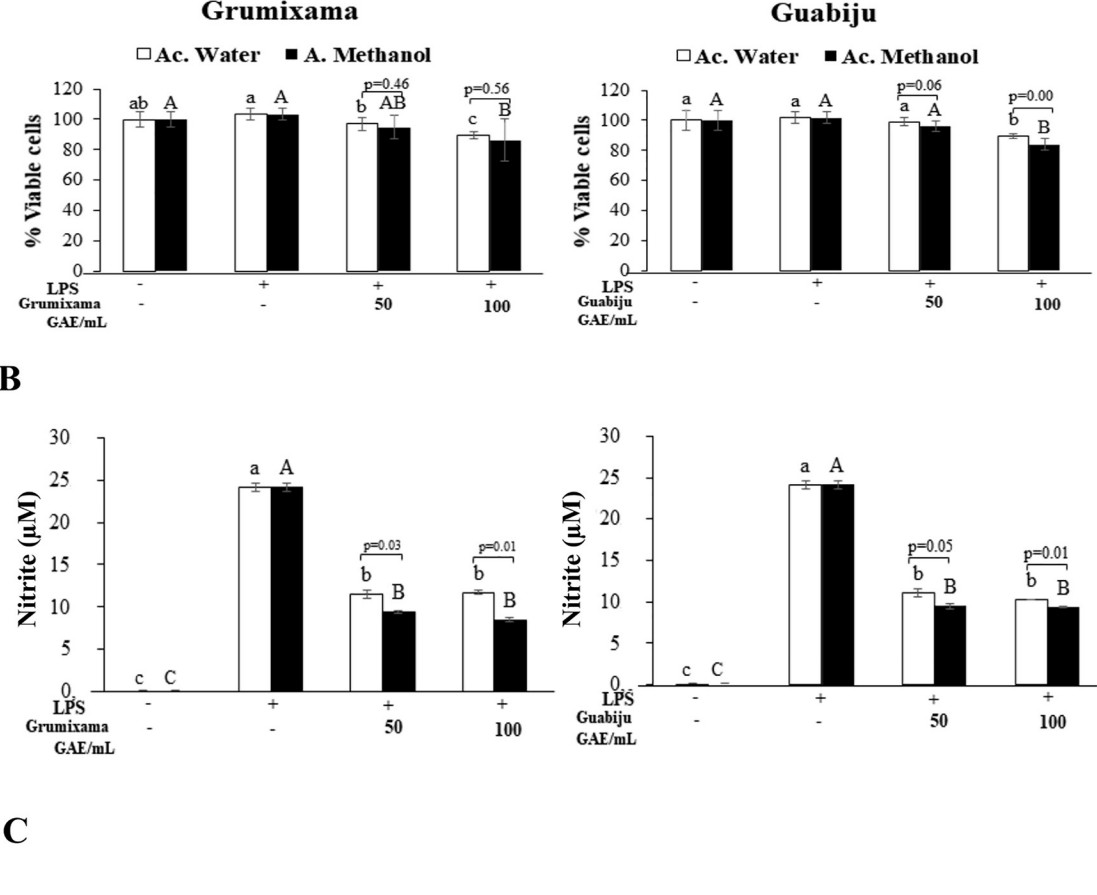

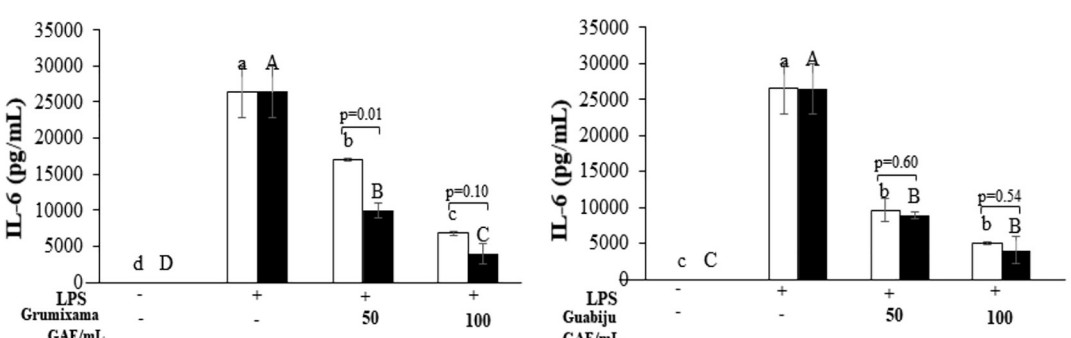

**Fig 3. Anti-inflammatory properties of grumixama and guabiju extracts obtained with different solvents. (A)** Effect of water or methanolic extracts from grumixama and guabiju of two different concentrations over the viability of RAW 264.7 macrophage, **(B)** Concentration of nitric oxide in the extracellular media of activated cells (LPS+), **(C)** Concentration of the pro-inflammatory cytokine IL-6 released in the extracellular media by activated cells (LPS+). Cells were induced into an inflammatory state (LPS+) in the presence of two different concentrations of extract. Non-activated cells (LPS-) were used as negative control. Results are presented as mean ± standard deviations. Different letters above columns indicate significant differences ($p < 0.05$) between the treatments using the same extract (lowercase letters: water extract, uppercase letters: methanol extract). Significant differences between the two extracts at the same concentration is indicated by the p-value above the respective pair of columns.

dependent response in which the methanolic extract produced 62% inhibition at the lowest concentration and near 85% inhibition at the highest concentration, while the water extracts achieved 36% and 74% inhibition, respectively. Interestingly, water and methanol extracts of guabiju were equally effective in inhibiting IL-6 release, with no significant differences between the two tested concentrations. All together, these results indicate that grumixama and guabiju phenolics extracts possess strong anti-inflammatory properties.

## Discussion

We investigated the phenolic composition of the Brazilian native fruits grumixama and guabiju, and their potential biological activity by testing their antioxidant and anti-inflammatory properties *in vitro*, comparing extractions performed using acidified water and acidified methanol. The quantification of phenolic groups (Table 1) indicated that these fruits are a good source of tannins and polyphenols, and at a lower extent of flavonoids, anthocyanins and 3-deoxyanthocyanins, with the methanolic extraction providing a higher yield. These results were in line with a previous study that reported higher levels of phenolic compounds extracted from different sorghum varieties using acidified ethanol, compared to those extracted with acidified water [17]. In another study carried out for the extraction of phenolic compounds from rambutan bark (*Nephelium lappaceum L*) with different solvents (water, ethanol and methanol) it was found that the extraction with methanol and ethanol resulted in the total phenolic compounds (TPC) 1.5 and 1.6 fold than those extracted with water, respectively [21]. In our current study, we observed that TPC in grumixama and guabiju extracts obtained with methanol were 2.2- and 1.4-fold, respectively, compared with those in the fruit extracts obtained with water. Previous study in the extraction of phenolic compounds have also reported that polar organic solvents were better than water to extract phenolic compounds such as phenolic acids and flavanoids [22]. This phenomenon might be due to a better interaction of organic solvent with the lyophilized fruit, releasing a greater quantity of compounds into the medium compared to the water. The extraction of phenolic compounds from plant materials is influenced by the solubility of the phenolic compounds in the extraction solvent which is affected by the solvent polarity [23]. Solvents with different polarities such as methanol and ethanol at different concentrations have been often used for extracting bioactive compounds from several plant matrices [13, 22].

In this study, the amounts of total phenolic compounds in grumixama and guabiju obtained by extraction with methanol were 546.6±14.93 and 399.98±15.8 mg GAE/100 g fresh fruit, respectively. Studies on whole fruits and pulp of grumixama harvested in different region in Brazil, reported TPC of 926.0±33.0 mg GAE 100g$^{-1}$ of fresh whole fruit (Xu et al., 2020), and 73.85 ± 3.21 mg GAE 100 g pulp$^{-1}$ [14]. Studies on guabiju have found TPC of 292 mg GAE/ 100 g fresh fruit and 2.43–4.42 mg GAE/g dry fruit [24]. Another study found TPC of 2.39 mg GAE/g dry fruit) [10]. When compared to other berries such as cranberry, blackberry, and strawberry, which contained TPC of 392.37, 42.2, and 40 mg GAE/100 g fruit, respectively [25], it can be inferred that both fruits, grumixama and guabiju, are considerably better sources of phenolic compounds.

The total anthocyanins content found in methanolic extracts of grumixama and guabiju were 108.2 and 140.2 mg C3G eq/100 g fresh fruit, respectively, while a previous study [12] reported 21.30 mg C3G eq/100 mL in grumixama juice. Literature reports indicate anthocyanins content ranging from 40–380 mg C3G eq/100 g in *Mahonia aquifolium* berry, with higher concentrations being obtained in ethanolic extracts than in water extracts [26]. Also, a previosu study reported the concentration of anthocyanins of 78–1058 mg/100 g fresh weight in black berries and 10–63 mg/100 g fresh weight in red berries [27]. These results indicated that

grumixama and guabiju are sources of anthocyanins comparable to other more commonly known purplish berries.

The levels of flavonoids extracted with methanol from grumixama and guabiju fruits were 50.9 and 44.8 mg quercetin eq/100 g fresh fruit, respectively. Flavonoids are responsible for a variety of biological process and particularly known for possessing antiviral activity. Tannins are by far the most abundant phenolic compound in guabiju and grumixama (Table 1). They are part of plant defense mechanisms, widely known for providing astringency to plant derived foods, but also for exhibiting diverse health benefits such as anticaries, antihistamine, anti-asthma, and antidiarrheal properties, as well as to have potential to cure intestinal infections and to prevent rhinitis [28].

The quantification of specific phenolics showed that while phenolic compounds whose structures are mainly composed of phenolic rings containing more than one hydroxyl group such as vanillic acid, ellagic acid and catechin, are highly abundant in both fruits, guabiju is particularly a better source of vanillic acid which contains a methyl group in its structure, and of coumaric, caffeic and ferulic acids, which are derivatives from the hydroxycinnamic acid and characterized for the presence of an aliphatic chains in their structure [29]. The better solubility of some phenolic compounds in methanol than in water, such as quercetin and luteolin could explain the higher concentration found in the methanolic extract.

A previous work [30] supports our findings in terms of the proportion of the different phenolic compounds present in guabiju, however, while this report indicates the presence of a considerable amount of syringic acid, in our study we did not find any peaks associated with this specific compound. Phenolic compounds are secondary metabolites in plants involved in different functions including reproduction, growth, pigmentation, defense against pathogens and protection from UV radiation, as such, their presence in plant and plant derived materials is highly dependent on the response of specialized cells to given environmental conditions. Then, these differences can be explained by potential variations in ripening state, soil composition, sun exposure, nutrients availability, and other environmental conditions specific to the geographical localization of the cultivars [30].

The ORAC values of grumixama and guabiju methanol extracts of 64.2 and 63.4 μmol Trolox eq/g fresh fruit are equivalent to 343.2. and 338.4 μmol Trolox eq/g dry fruit, respectively. Previous study on grumixama pulp and seed reported the ORAC values of 134.28 and 111.11 μmol Trolox eq/g dry fruit, respectively [31]. A previous work [8] on the whole grumixama fruits using ethanol/water (80/20) found ORAC value of 477.45 μmol Trolox eq/ g dry fruit. Another study conducting ORAC assay for several freeze-dried fruits including blueberry, cherry, cranberry and strawberry reported 392.25, 273.75, 444.50, and 412.5 μmol Trolox eq/g dry fruit, respectively, [32]. The ORAC values of grumixama and guabiju in the current study were in proximity to those of berries which are widely consumed in the world indicating a highly potential use of the Brazilian native fruits grumixama and guabiju as antioxidant sources in the future.

The fact that no significant differences were observed between extraction solvents for DPPH or NO scavenging activity suggest that the compounds mainly responsible for these activities are equally extracted in water and methanol, while the significantly higher ORAC capacity observed in methanolic extracts implies that the compounds involved in the oxygen radical absorbance capacity have higher affinity for methanol than for water. We used three different *in vitro* assays to determine the antioxidant activity of grumixama and guabiju acidic extracts. DPPH scavenging activity measured the reducing ability of the extracts based mainly on the electron transfer reaction, and considered rapid, simple, and highly sensitive [33]. NO scavenging is based on the spontaneous generation of NO by sodium nitroprusside in aqueous solution upon interaction with oxygen and in the presence of the

extracts that competes with oxygen resulted in the reduced production of NO [34]. On the other hand, ORAC assay is a hydrogen atom transfer-based assay where the generated peroxyl radical is measured which can provide the ability of the antioxidant extracts to break radical chain formation [35]. Previous studies have reported that the number of OH in the phenolic compounds has a good correlation with radical scavenging capacity measured by DPPH but not by ORAC, which in contrast is more favored by OH substitutions, particularly in the A and B rings of flavonoids [36]. This premise allows to hypothesize that water but not methanol soluble compounds not identified here might be contributing to DPPH scavenging activity, making it statistically equal for both extracts, while specific phenolics and particularly flavonoids with a reduced presence of OH groups preferably present in the methanolic extract might be contributing to the ORAC activity. In addition, slight variations in the results found when compared from previous studies can be attributed to differences in extraction protocols (such as time used, reagents, and equipment efficiency), sample storage protocols, and also to differences in analysis parameters and methodology used.

When injury or infection occurs, microbial endotoxins such as LPS trigger signaling cascades in immune cells, primarily macrophages, that result in the production and release of antimicrobial products, such as nitric oxide, and pro-inflammatory cytokines (mainly TNF-α, IL-6, IL-1β), initiating the acute inflammatory response, which prevent further damage and ultimately leads to healing and restoring of tissue function [37]. However, when the inflammatory response becomes aberrant and dysregulated, it can cause damage to the host and cause degenerative diseases such as asthma, diabetes, arthritis, inflammatory bowel disease, cancer, among others. Therefore, there is a constant search for mechanism that facilitate the regulation of the inflammatory response. In this study, we evaluated the performance of grumixama and guabiju extracts in an *in vitro* model of inflammation.

NO is a signaling molecule involved in the regulation of muscle and airway tone, insulin secretion, and intestine peristalsis; this highly reactive molecule can react with cell proteins and impair their function, acting in a non-selective manner against pathogens, but also producing damage to the host cells [38]. IL-6 is produced by senescent cells, contributing to senescence-induced inflammation and being further involved in age-dependent pathologies, but it is also produced by immune cells and acts as an amplifier of inflammation via the synergistic interaction with the signal transducer and activator of transcription 3 (STAT3) pathway and the nuclear factor-kappa B (NF-κB) pathway, promoting the further production of IL-6 and other pro-inflammatory cytokines and chemokines. Our results indicate that both fruits can exhibit anti-inflammatory properties by significantly decreasing the release of NO and the pro-inflammatory cytokine IL-6.

While no significant differences were found between water and methanol extracts for the NO radical scavenging activity, the cellular model showed that the methanolic extract was significantly more effective reducing the level of NO released to the extracellular environment. This suggests that while the compounds present in both extracts are equally effective at scavenging NO radicals, specific compounds only present in the methanolic extract can reduce NO production. The research by Lazarini et al. [39] supports our findings on grumixama fruit anti-inflammatory activity by demonstrating the ability of the ethanolic extract to decrease the activation of NF-κB pathway in murine macrophages and reduce the release of TNF-α and the influx of immune cells (neutrophils) to the peritoneal cavity in an *in vivo* model (mice) of carrageenan-induced paw edema [39]. Other parts of the grumixama plant, such as the leaves, have been also demonstrated to exert anti-inflammatory properties [40]. Up to date and to the best of our knowledge, our study is the first report presenting the potential anti-inflammatory properties of guabiju extracts.

## Conclusions

Our study showed that phenolic compounds from grumixama and guabiju possess antioxidant and anti-inflammatory properties. The results can be used to promote the consumption of these fruits and highlight the potential health benefits of grumixama and guabiju.

## Supporting information

**S1 File. Minimal data set.**
(XLSX)

## Acknowledgments

To the growers of the native fruits of Rio Grande do Sul: Rosangela, Dorlei, Sérgio and Ilgo for providing the fruits for the research.

## Author Contributions

**Conceptualization:** Bruna Tischer, Vermont P. Dia.

**Data curation:** Bruna Tischer, Vermont P. Dia.

**Formal analysis:** Bruna Tischer, Philipus Pangloli, Vermont P. Dia.

**Funding acquisition:** Bruna Tischer, Vermont P. Dia.

**Investigation:** Bruna Tischer, Philipus Pangloli, Andrea Nieto-Veloza, Matthew Reeder.

**Methodology:** Bruna Tischer, Philipus Pangloli, Andrea Nieto-Veloza, Vermont P. Dia.

**Project administration:** Vermont P. Dia.

**Resources:** Bruna Tischer, Vermont P. Dia.

**Supervision:** Vermont P. Dia.

**Validation:** Bruna Tischer, Philipus Pangloli, Vermont P. Dia.

**Visualization:** Bruna Tischer, Philipus Pangloli, Vermont P. Dia.

**Writing – original draft:** Bruna Tischer, Philipus Pangloli.

**Writing – review & editing:** Bruna Tischer, Philipus Pangloli, Andrea Nieto-Veloza, Matthew Reeder, Vermont P. Dia.

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
