## [Decision Letter · Decision Letter 0]

8 Dec 2022

PONE-D-22-26448Bioactive compounds, antioxidant capacity and anti-inflammatory activity of native fruits from BrazilPLOS ONE

Dear Dr. Dia,

Thank you for submitting your manuscript to PLOS ONE. After careful consideration, we feel that it has merit but does not fully meet PLOS ONE’s publication criteria as it currently stands. Therefore, we invite you to submit a revised version of the manuscript that addresses the points raised during the review process.

We look forward to receiving your revised manuscript.

Kind regards,

Umakanta Sarker

Academic Editor

PLOS ONE

Journal Requirements:

“No”

5. Please upload a new copy of Figure 1 as the detail is not clear. Please follow the link for more information: " ext-link-type="uri" xlink:type="simple">https://blogs.plos.org/plos/2019/06/looking-good-tips-for-creating-your-plos-figures-graphics/"
https://blogs.plos.org/plos/2019/06/looking-good-tips-for-creating-your-plos-figures-graphics/

Reviewers' comments:

Reviewer's Responses to Questions

**Comments to the Author**

1. Is the manuscript technically sound, and do the data support the conclusions?

Reviewer #1: Yes

Reviewer #2: Yes

2. Has the statistical analysis been performed appropriately and rigorously? 

Reviewer #1: Yes

Reviewer #2: Yes

3. Have the authors made all data underlying the findings in their manuscript fully available?

Reviewer #1: Yes

Reviewer #2: Yes

4. Is the manuscript presented in an intelligible fashion and written in standard English?

Reviewer #1: Yes

Reviewer #2: Yes

5. Review Comments to the Author

Reviewer #1: 1. In Abstract, kindly put the values from the results into the abstract and also show the comparison, abstract should be re-write.

2. In Introduction, authors must declassify the native fruit names (including common and scientific names).

3. In Materials and Methods section, in DPPH method reference is missing.

4. Conclusion should be re-write.

5. Figure 1 and 3 quaility is too poor.

Reviewer #2: This manuscript has reasonable data to support the conclusions. However, there are some issues that should be clarified or revised as follows:

1. In the introduction part, the reasons should be added for the selection of various standard substances for HPLC quantitative analysis.

2. In the result part, the results of radicals scavenging and anti-inflammatory effects should be presented in the form of percent inhibition for easier comparison with other studies.

3. In the discussion part:

(1) The authors should detail how the differences of three methods for testing antioxidant activity performed in this study in terms of antioxidant mechanisms were.

(2) In order to compare the concentrations of active substances in both plants from this study with those found in other plants in the previous studies, the variation of the extraction methods and testing conditions must take into account.

4. The typing errors should be corrected throughout the manuscript.

5. The image quality in the manuscript should be improved.

6. PLOS authors have the option to publish the peer review history of their article (what does this mean?). If published, this will include your full peer review and any attached files.

Reviewer #1: No

Reviewer #2: No

---

## [Author Response · Author response to Decision Letter 0]

4 Jan 2023

Journal Requirements:

Response. The manuscript has been formatted according to PLOS ONE’s style requirements.

Response. This has been updated.

“No”

Response. Done.

Response. The minimal data set has been uploaded as Supporting Information file.

5. Please upload a new copy of Figure 1 as the detail is not clear. Please follow the link for more information: https://blogs.plos.org/plos/2019/06/looking-good-tips-for-creating-your-plos-figures-graphics/" https://blogs.plos.org/plos/2019/06/looking-good-tips-for-creating-your-plos-figures-graphics/

Response. This has been re-uploaded with better quality and higher resolution.

Reviewers' comments:

Reviewer's Responses to Questions

Comments to the Author

1. Is the manuscript technically sound, and do the data support the conclusions?

Reviewer #1: Yes

Reviewer #2: Yes

2. Has the statistical analysis been performed appropriately and rigorously? 

Reviewer #1: Yes

Reviewer #2: Yes

3. Have the authors made all data underlying the findings in their manuscript fully available?

Reviewer #1: Yes

Reviewer #2: Yes

4. Is the manuscript presented in an intelligible fashion and written in standard English?

Reviewer #1: Yes

Reviewer #2: Yes

5. Review Comments to the Author

Reviewer #1: 1. In Abstract, kindly put the values from the results into the abstract and also show the comparison, abstract should be re-write.

Response. The Abstract has been re-written per reviewer’s recommendations. Thank you.

2. In Introduction, authors must declassify the native fruit names (including common and scientific names).

Response. This has been highlighted in page 3, lines 68-70.

3. In Materials and Methods section, in DPPH method reference is missing.

Response. This is reference 18 as highlighted in pages 7-8, lines 173-176. 

4. Conclusion should be re-write.

Response. This has been re-written to give a broad and general conclusion of the findings without repeating the results section.

5. Figure 1 and 3 quaility is too poor.

Response. These figures were modified and re-uploaded with better quality and high resolution.

Reviewer #2: This manuscript has reasonable data to support the conclusions. However, there are some issues that should be clarified or revised as follows:

1. In the introduction part, the reasons should be added for the selection of various standard substances for HPLC quantitative analysis.

Response. The following sentences were added in the Introduction. Thank you.

Phenolic compounds were both quantified spectrophotometrically and individual compounds were identified and quantified by high performance liquid chromatography (HPLC) using different analytical standards. These standards were chosen based on availability as well as those reported in the literature to be present in these fruits. 

2. In the result part, the results of radicals scavenging and anti-inflammatory effects should be presented in the form of percent inhibition for easier comparison with other studies.

Response. We respectfully take this comment into consideration. We believe that the current presentation of results as actual concentrations of each radicals (DPPH and NO) and actual concentrations of inflammatory markers (NO and IL-6) are better since we are using different concentrations of the extracts. Moreover, these are clearly delineated in Table 3 footnote (DPPH and NO) and Figure 3 legend (NO and IL-6) as relative to blank (100%) and positive control (LPS+), respectively. Thank you.

3. In the discussion part:

(1) The authors should detail how the differences of three methods for testing antioxidant activity performed in this study in terms of antioxidant mechanisms were.

Response. The following sentences were added in the Discussion section. Thank you.

We used three different in vitro assays to determine the antioxidant activity of grumixama and guabiju acidic extracts. DPPH scavenging activity measured the reducing ability of the extracts based mainly on the electron transfer reaction, and considered rapid, simple, and highly sensitive [34]. NO scavenging is based on the spontaneous generation of NO by sodium nitroprusside in aqueous solution upon interaction with oxygen and in the presence of the extracts that competes with oxygen resulted in the reduced production of NO [35]. On the other hand, ORAC assay is a hydrogen atom transfer-based assay where the generated peroxyl radical is measured which can provide the ability of the antioxidant extracts to break radical chain formation [36]. 

The following references were added:

34. Moon, J. K., Shibamoto, T. Antioxidant assays for plant and food components. Journal of Agricultural and Food Chemistry, 2009, 57, 1655–1666. 

35. Sarwar, R., Farooq, U., Khan, A., Naz, S., Khan, S., Khan, A., Rauf, A., Bahadar, H., Uddin, R. Evaluation of antioxidant, free radical scavenging, and antimicrobial activity of Quercus incana Roxb. Frontiers in Pharmacology, 2015, 6, 277.

36. Huang, D., Ou, B., Prior, R. L. The chemistry behind antioxidant capacity assays. Journal of Agricultural and Food Chemistry,

2005, 53, 1841–1856.

(2) In order to compare the concentrations of active substances in both plants from this study with those found in other plants in the previous studies, the variation of the extraction methods and testing conditions must take into account.

Response. The following sentences were added in the Discussion section. Thank you.

In addition, slight variations in the results found when compared from previous studies can be attributed to differences in extraction protocols (such as time used, reagents, and equipment efficiency), sample storage protocols, and also to differences in analysis parameters and methodology used.

4. The typing errors should be corrected throughout the manuscript.

Response. The manuscript has been proof-read to correct typing errors.

5. The image quality in the manuscript should be improved.

Response. The images were re-uploaded with better quality and higher resolution.

6. PLOS authors have the option to publish the peer review history of their article (what does this mean?). If published, this will include your full peer review and any attached files.

Do you want your identity to be public for this peer review? For information about this choice, including consent withdrawal, please see our Privacy Policy.

Reviewer #1: No

Reviewer #2: No

---

## [Decision Letter · Decision Letter 1]

3 Apr 2023

PONE-D-22-26448R1Bioactive compounds, antioxidant capacity and anti-inflammatory activity of native fruits from BrazilPLOS ONE

Dear Dr. Dia,

Thank you for submitting your manuscript to PLOS ONE. After careful consideration, we feel that it has merit but does not fully meet PLOS ONE’s publication criteria as it currently stands. Therefore, we invite you to submit a revised version of the manuscript that addresses the points raised during the review process.

We look forward to receiving your revised manuscript.

Kind regards,

Umakanta Sarker

Academic Editor

PLOS ONE

Journal Requirements:

Reviewers' comments:

Reviewer's Responses to Questions

**Comments to the Author**

1. If the authors have adequately addressed your comments raised in a previous round of review and you feel that this manuscript is now acceptable for publication, you may indicate that here to bypass the “Comments to the Author” section, enter your conflict of interest statement in the “Confidential to Editor” section, and submit your "Accept" recommendation.

Reviewer #1: All comments have been addressed

Reviewer #3: All comments have been addressed

2. Is the manuscript technically sound, and do the data support the conclusions?

Reviewer #1: Partly

Reviewer #3: Yes

3. Has the statistical analysis been performed appropriately and rigorously? 

Reviewer #1: Yes

Reviewer #3: Yes

4. Have the authors made all data underlying the findings in their manuscript fully available?

Reviewer #1: Yes

Reviewer #3: Yes

5. Is the manuscript presented in an intelligible fashion and written in standard English?

Reviewer #1: Yes

Reviewer #3: Yes

6. Review Comments to the Author

Reviewer #1: Please make conclusion more scientific as per results obtained. It is not a review but research, so pay attention.

Reviewer #3: the revised version reports all the corrections required by the Reviewers. Now the paper can be accepted

7. PLOS authors have the option to publish the peer review history of their article (what does this mean?). If published, this will include your full peer review and any attached files.

Reviewer #1: No

Reviewer #3: No

---

## [Author Response · Author response to Decision Letter 1]

3 Apr 2023

April 3, 2023

Umakanta Sarker

Academic Editor

PLOS ONE

Dear Dr. Sarker,

Thank you very much for orchestrating the review of our manuscript entitled “Bioactive compounds, antioxidant capacity and anti-inflammatory activity of native fruits from Brazil” for potential publication in PLOS One. This study involved several Brazilian native fruit producers, fruit researchers at the Federal University of Rio Grande do Sul, Brazil and the University of Tennessee, United States. These native fruits have great potential for application as a source of bioactive compounds with very promising antioxidant and anti-inflammatory properties. It is known that there is a great diversity of native fruits in Brazil, however there is a lack of more advanced studies in the area of detailing the chemical composition and pharmacological activities, and these are extremely important to stimulate the production chain in this sector, so that greater investment occurs in the area and also application of these vegetable sources in food products.

Our study aims to disseminate these results for grumixama and guabiju fruits, of which there is little material published in the literature. We believe that with these results we will be able to help stimulate the production, consumption, and production chain of native Brazilian fruits.

The figures were uploaded and checked by PACE. Below, you will find the point-by-point response to the reviews and suggestions made by the reviewers and PLOS One editorial team. The minimal set has been uploaded as Supplementary Material file. We are also uploading both marked and unmarked version of the manuscript.

Thank you very much for considering this work for publication.

Sincerely,

VERMONT P DIA, Ph D

Associate Professor

Department of Food Science

The University of Tennessee

2510 River Dr Knoxville TN 37996

vdia@utk.edu, 865-974-7265

Journal Requirements:

Response. The reference list has been checked. None of the cited papers was retracted.

Reviewers' comments:

Reviewer's Responses to Questions

Comments to the Author

1. If the authors have adequately addressed your comments raised in a previous round of review and you feel that this manuscript is now acceptable for publication, you may indicate that here to bypass the “Comments to the Author” section, enter your conflict of interest statement in the “Confidential to Editor” section, and submit your "Accept" recommendation.

Reviewer #1: All comments have been addressed

Reviewer #3: All comments have been addressed

Response. Thank you very much.

2. Is the manuscript technically sound, and do the data support the conclusions?

Reviewer #1: Partly

Response. The conclusion has been revised per suggestion in Review Comments to the Author.

Reviewer #3: Yes

Response. Thank you very much.

3. Has the statistical analysis been performed appropriately and rigorously? 

Reviewer #1: Yes

Reviewer #3: Yes

Response. Thank you very much.

4. Have the authors made all data underlying the findings in their manuscript fully available?

Reviewer #1: Yes

Reviewer #3: Yes

Response. Thank you very much.

5. Is the manuscript presented in an intelligible fashion and written in standard English?

Reviewer #1: Yes

Reviewer #3: Yes

Response. Thank you very much.

6. Review Comments to the Author

Reviewer #1: Please make conclusion more scientific as per results obtained. It is not a review but research, so pay attention.

Response. The conclusion has been revised according to the results obtained and read as follows:

Our study showed that phenolic compounds from grumixama and guabiju possess antioxidant and anti-inflammatory properties. The results can be used to promote the consumption of these fruits and highlight the potential health benefits of grumixama and guabiju.

Reviewer #3: the revised version reports all the corrections required by the Reviewers. Now the paper can be accepted

Response. Thank you very much.

7. PLOS authors have the option to publish the peer review history of their article (what does this mean?). If published, this will include your full peer review and any attached files.

Do you want your identity to be public for this peer review? For information about this choice, including consent withdrawal, please see our Privacy Policy.

Reviewer #1: No

Reviewer #3: No

Response. Thank you very much.

---

## [Decision Letter · Decision Letter 2]

27 Apr 2023

Bioactive compounds, antioxidant capacity and anti-inflammatory activity of native fruits from Brazil

PONE-D-22-26448R2

Dear Dr. Dia,

We’re pleased to inform you that your manuscript has been judged scientifically suitable for publication and will be formally accepted for publication once it meets all outstanding technical requirements.

Kind regards,

Umakanta Sarker

Academic Editor

PLOS ONE

Additional Editor Comments (optional):

Reviewers' comments:

Reviewer's Responses to Questions

**Comments to the Author**

1. If the authors have adequately addressed your comments raised in a previous round of review and you feel that this manuscript is now acceptable for publication, you may indicate that here to bypass the “Comments to the Author” section, enter your conflict of interest statement in the “Confidential to Editor” section, and submit your "Accept" recommendation.

Reviewer #1: All comments have been addressed

2. Is the manuscript technically sound, and do the data support the conclusions?

Reviewer #1: Yes

3. Has the statistical analysis been performed appropriately and rigorously? 

Reviewer #1: Yes

4. Have the authors made all data underlying the findings in their manuscript fully available?

Reviewer #1: Yes

5. Is the manuscript presented in an intelligible fashion and written in standard English?

Reviewer #1: Yes

6. Review Comments to the Author

Reviewer #1: Authors have addressed and resolved all the issues very well. Now the qualtiy of manuscript is up to the mark.

7. PLOS authors have the option to publish the peer review history of their article (what does this mean?). If published, this will include your full peer review and any attached files.

Reviewer #1: No

---

## [Editor Report · Acceptance letter]

28 Apr 2023

PONE-D-22-26448R2 

Bioactive compounds, antioxidant capacity and anti-inflammatory activity of native fruits from Brazil 

Dear Dr. Dia:

I'm pleased to inform you that your manuscript has been deemed suitable for publication in PLOS ONE. Congratulations! Your manuscript is now with our production department. 

Kind regards, 

on behalf of

Professor Umakanta Sarker 

Academic Editor

PLOS ONE